# Numerical modeling vs experiment of formic acid and formate ion behavior under gamma radiation at several pH values: Implications on prebiotic chemistry

**Alejandro Paredes-Arriaga**[1,2]*, **Anayelly López-Islas**[1], **Diego Frias**[3], **Ana Leonor Rivera**[1,4], **Guadalupe Cordero-Tercero**[5], **Sergio Ramos-Bernal**[1], **Alicia Negrón-Mendoza**[1]*

**1** Instituto de Ciencias Nucleares, Universidad Nacional Autónoma de México, Circuito Exterior s/n, Ciudad Universitaria, Coyoacán, CDMX, México, **2** Posgrado en Ciencias de la Tierra, Universidad Nacional Autónoma de México, Circuito Exterior s/n, Ciudad Universitaria, Coyoacán, CDMX, México, **3** Departamento de Ciências Exatas e da Terra, Universidade do Estado da Bahia (UNEB), Silveira Martins, Salvador, BA, Brazil, **4** Centro de Ciencias de la Complejidad, Universidad Nacional Autónoma de México, Circuito Exterior s/n, Ciudad Universitaria, Coyoacán, CDMX, México, **5** Instituto de Geofísica, Universidad Nacional Autónoma de México, Circuito Exterior s/n, Ciudad Universitaria, Coyoacán, CDMX, México

\* alejandro.paredes@correo.nucleares.unam.mx (APA); negron@nucleares.unam.mx (ANM)

**Data Availability Statement:** Data included in article/supplementary material/referenced in article.

## Abstract

Formic acid is consistently produced and detected in prebiotic chemistry experiments, constituting a precursor of many carboxylic acids and amino acids. Its behavior with exposure to gamma radiation varies with the pH and solution concentration. This work aimed to model different environmental conditions for formic acid under ionizing radiation using a system of coupled differential equations based on chemical kinetics. An ensemble of radiolysis reaction mechanisms was generated for formic acid at pH 1.5 and formate ion at pH 9, both with radiation doses from 0 to 2 kGy. This was also done for systems with both species in equilibrium, using high molar concentrations, long irradiation times, and large irradiation doses (from 0 to 70 kGy). The results show that these systems can be modeled with a high statistical relationship between the computed solutions and the experimental data; furthermore, the synthesis and degradation of the radiolysis products can be followed. Another dimension of the issue of prebiotic environments was explored using ionizing radiation and analyzing the reactions at various pH values (acidic to basic media). These models allow one to gain insights into the behavior of molecules that are difficult to detect or analyze in the laboratory. Additionally, they offer the possibility of studying potential prebiotic environments.

## 1 Introduction

Formic acid (HCOOH) is an organic molecule commonly found in the interstellar medium [1], along with other molecules that can be considered essential building blocks of biomolecules, such as methanol ($CH_3OH$), acetic acid ($CH_3COOH$), acetamide ($CH_3CONH_2$), and glycine ($C_2H_5NO_2$), among others [2]. In addition, formic acid has been detected in various

Model developed in this work is available at github: https://github.com/A-Paredes-Arriaga/Chemical-kinetics_EDO, DOI: 10.5281/zenodo.13966974

**Funding:** The author(s) received no specific funding for this work.

comets, such as 67P/Churyumov–Gerasimenko [1], C/2013 R1-Lovejoy [3], and C/1995 O1-Hale-Bopp [4]; the protoplanetary disk of TW Hydrae [5]; and meteorites, such as Murchinson, Allende, Parnallee, Leedey, and Abee [6].

Formic acid and other simple acids can be formed via the radiolysis of carbonates in hydrothermal systems [7, 8], by electric discharges in volcanic dust clouds, and from asteroid impacts [9], all of which are typical scenarios for chemical evolution. Formic acid and formaldehyde are carbonylic compounds often produced in prebiotic simulation experiments under reducing and oxidizing conditions [9–12].

On the geological timescale, the prebiotic era is limited to the time period from around 4.2 to 3.8 Gya ago [13, 14]. The existence of a variety of environments on primitive Earth that could be linked to chemical evolution processes necessitates conducting experiments under different conditions–by varying the pH, pressure, and temperature, among other variables [15]. Ionizing radiation constitutes one of the energy sources that promote the synthesis and degradation of organic molecules [16–18]. The primary contributors to this form of energy on primitive Earth were the radionuclides $^{40}K$, $^{238}U$, $^{235}U$, and $^{232}Th$ [19–21]. The experiments presented in this work explore the irradiation of carboxylic acid under various pH conditions, concentrations, and dose intensities. These conditions could be present in environments such as acidic lakes [22], alkaline lakes [23], parts of hot spring systems [24, 25], shallow fresh water bodies [26, 27], and oceans [28, 29]. In particular, pH variations in prebiotic simulations are a common feature when subjecting a system to energy sources such as radiation or heat [10, 12, 30–32]. This represents a priority problem to investigate because the solution pH can affect the reactions of a system and, the resulting products [33].

The radiolysis of formic acid has formed the subject of various studies [34–37]. Different reaction mechanisms have been proposed at different pH values [38–40] and concentrations [41] and under oxygen [38] and oxygen-free [42] conditions. In addition, as a reducing agent, formic acid can convert some hydroxyamino acids to their reduced amino acid forms [11]. Formic acid exist in an acid-base equilibrium with the formate ion, $HCOO^-$[43]; with formic acid dominating at pH $\leq$ 1.5 and the formate ion dominating at pH $\geq$ 6 (Fig 1).

This work aimed to study the behavior of formic acid, the formate ion and their products under a high ionizing radiation field and different pH conditions and concentrations to simulate prebiotic environments, using a mathematical model to reproduce the experimental data. Given that formic acid, the formate ion, and their mixture entail different sets of reactions (and reaction rate constants), each system constitutes a separate problem of study, relevant to a complex prebiotic system with a variation of its initial conditions. Experimental data from previous studies have been used to validate our numerical models. The study concludes with an inter- and multidisciplinary perspective, addressing problems related to chemical evolution, chemical kinetics, radiation chemistry, differential equations, and computer modeling.

## 2 Methods

We develop models with different conditions of pH, radiation dose, and concentrations (Table 1); with the aim of modeling the physico-chemical conditions of different probable prebiotic environments as described in the Introduction section. In addition, some of these specific conditions have been determined by experiments previously published by other authors that we can adapt to a prebiotic environment.

### 2.1 Numerical modeling

We developed numerical models for four chemical systems, all under oxygen-free and room temperature conditions: (1) formic acid at pH 1.5, concentration = $1*10^{-3}$ mol/L, maximum

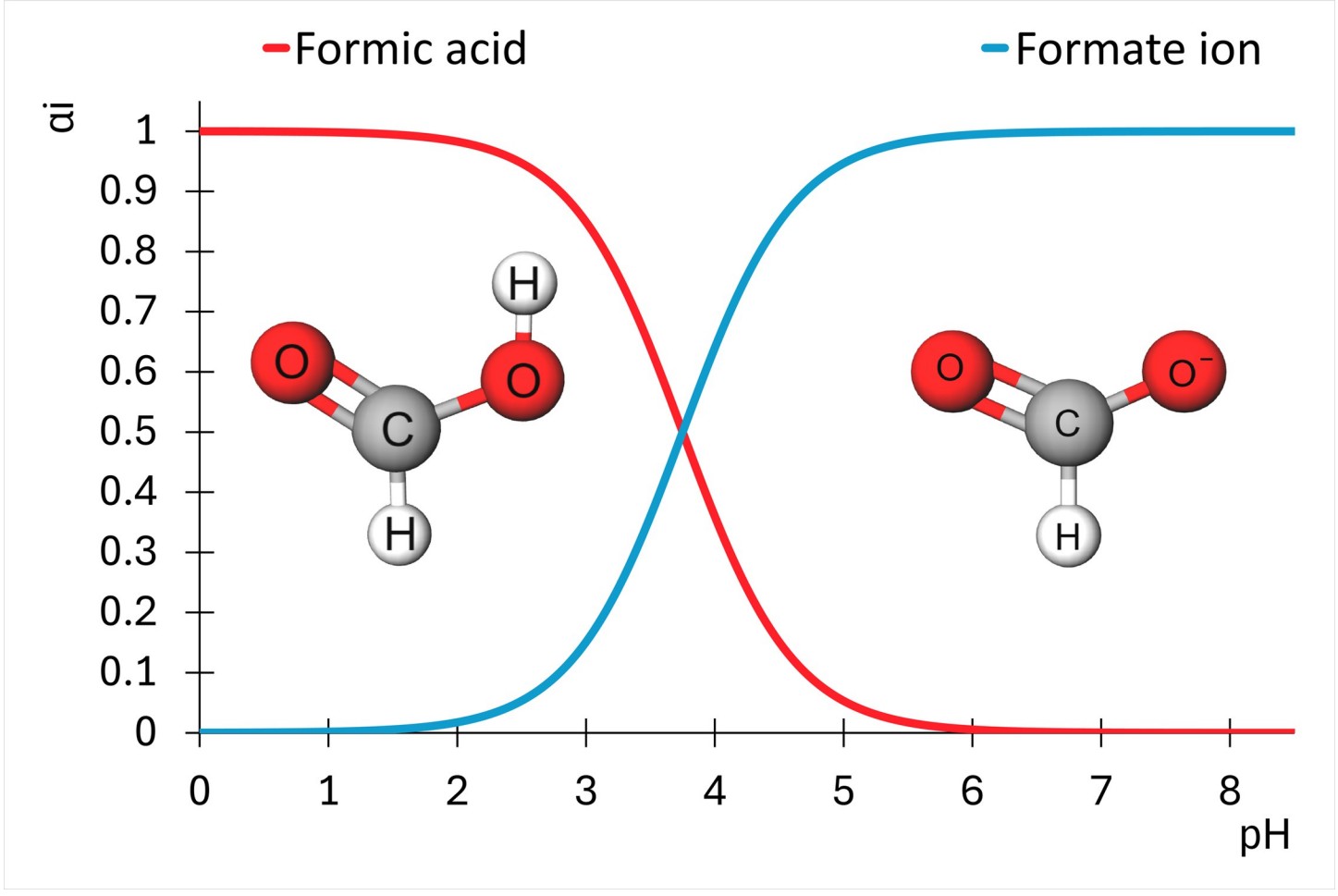

**Fig 1. Species distribution diagram of formic acid and formate as a function of pH.**

irradiation dose = 2 kGy, and dose intensity = 0.90 Gy/min; (2) formate at a pH 9, concentration = $1*10^{-3}$ mol/L, maximum irradiation dose = 2 kGy, and dose intensity = 0.90 Gy/min; (3) formic acid and formate mixture at pH 2, respective concentrations = $1.96*10^{-2}$ mol/L and $0.04*10^{-2}$ mol/L, maximum irradiation dose = 70 kGy, and dose intensity = 266 Gy/min; (4) formic acid and formate mixture at pH 3.75, concentrations of both = $1*10^{-2}$ mol/L respectively, maximum irradiation dose = 90 kGy, and dose intensity = 266 Gy/min. We also

**Table 1. Chemical systems modeled in this work.** Each model is in oxygen-free conditions and room temperature.

| Reactants | pH | Concentration [mol/L] | Maximum irradiation dose [kGy] | Dose intensity [Gy/min] | Experimental data |
|---|---|---|---|---|---|
| Formic acid | 1.5 | $1*10^{-3}$ | 2 | 0.90 | [42] |
| Formate ion | 9 | $1*10^{-3}$ | 2 | 0.90 | [42] |
| Formic acid and formate ion | 2 | $1.96*10^{-2}$ and $0.04*10^{-2}$ (respectively) | 70 | 266 | This work |
| Formic acid and formate ion | 3.75 | $1*10^{-2}$ (both) | 90 | 266 | Only theoretical model |
| Formic acid and formate ion | 3.75 | $1*10^{-3}$ (both) | 2 | 0.90 | Only theoretical model |

varied the initial concentration and maximum irradiation dose from the minimum values of $1*10^{-3}$ mol/L and 2 kGy, respectively (see Table 1).

The numerical models for formic acid (pH 1.5) and formate (pH 9) were compared with experimental data from the study by Horne et al. (2020). The numerical model at pH 2 was compared with experimental data obtained with the setup presented in Section 2.2. Further, we proposed a reaction mechanism for each chemical system in Table 1 based on a literature review.

Each chemical reaction system was written as a set of coupled differential equations, and each chemical species in a system was represented by one equation [44, 45]. The Eq 1 includes information about the molecules formed, the molecules that decay, and a source term ($f_i$) simulating the radiation source [46, 47]:

$$\frac{dX_i(t)}{dt} = f_i + \sum_{m=0}^{N} \sum_{n=0}^{N} k_{m,n}^{(i)} X_m(t) X_n(t) - X_i(t) \sum_{j\neq0}^{N} k_{i,j}^{(i)} X_j(t), \tag{1}$$

where $X_i(t)$, $X_j(t)$, $X_m(t)$, and $X_n(t)$, is the molar concentration of chemical species $i$, $j$, $m$ and $n$, at time $t$; $k_{i,j}^{(i)}$ is the rate constant of the reaction between the species $i$ and $j$, $k_{m,n}^{(i)}$ is the rate constant for the reaction between chemical species $m$ and $n$ that produced the $i$ species. $\frac{dX_i(t)}{dt}$ is the change in the molar concentration of species $i$ at time $t$. The positive part of the sums represents the formation of the species $i$ as a result of the reaction between the molecules $m$ and $n$ at a rate $k_{m,n}^{(i)}$. The negative part of the sums represents the degradation of the species $i$ by the reaction of the molecules $i$ and $j$ at a rate $k_{i,j}^{(i)}$. $f_i$ is an external energy source, which in this case is the gamma radiation source.

The source term ($f_i$) is given by the Eq 2 [45, 47, 48]:

$$f_i(I_d) = \frac{6.2*10^{11}}{3.6\ N_A} \frac{M_i}{M_{H_2O}} G_i \left[ I_d * \left( 6*10^3 \right) \right], \tag{2}$$

where $N_A$ is Avogadro's number ($6.022*10^{23}$ molecules), $M_i$ is the molecular mass of species $i$, $M_{H_2O}$ is the molecular mass of water (18.02 g/mol), $G_i$ is the radiochemical yield of species $i$ when the system absorbs 100 eV, and $I_d$ is the dose rate (Gy/min).

Each coupled nonlinear differential equation system was solved building a *Python (3.7.9)* code, employing the *solve.ivp* function from the *numpy* library and implementing *BDF*, an implicit multistep method of variable order (1 to 5) utilizing a backward differentiation formula for the derivative approximation [49]. The general algorithm and full code are available at https://github.com/A-Paredes-Arriaga/Chemical-kinetics_EDO [50]. Finally, the statistical analysis included calculations of the root mean square error (RMSE), $R^2$ value, and standard deviation and a quantile-quantile (Q-Q) plot of the residuals.

## 2.2 Experimental setup

**2.2.1 Preparation and irradiation of formic acid solution.** Formic acid (0.02 mol/L) was prepared using a formic acid solution (95% pure, Sigma-Aldrich, Saint Louis, Missouri, USA) and triple distilled water. The samples were degassed with argon for 15 minutes and then irradiated with gamma rays ($^{60}$Co source, Gamma-beam 651 PT at the Instituto de Ciencias Nucleares, UNAM). Formic acid aliquots (5 mL, pH 2) were exposed to doses of gamma radiation from 0 to 70 kGy.

**2.2.2 Determination of formic acid degradation by titration.** The decomposition of formic acid was measured by titration after exposure to gamma radiation. Sodium hydroxide

(0.02 mol/L) was used as the titrant, 0.5% phenolphthalein (Sigma-Aldrich) as the indicator, and formic acid ($1*10^{-3}$ mol/L) as the standard.

**2.2.3 Quantification of carbon dioxide from formic acid solutions after irradiation.** A combined ion-selective electrode, ISE (Orion ™ 9502BNWP), was used for the detection of carbon dioxide ($CO_2$) in the formic acid solutions. Standard $CaCO_3$ solutions ($1*10^{-2}$ and $1*10^{-4}$ mol/L) were prepared to calibrate the electrode. Subsequently, 5 mL of a $CO_2$ buffer solution was added to 50 mL of each solution.

## 3 Results

### 3.1 Formic acid at pH 1.5

The reaction mechanism for formic acid at pH 1.5 is represented by Reactions 1.1 to 1.8, based on different studies detailing various possibles routes (Table 2). At a low pH, the solvated electron and aqueous hydrogen react to generate H• radicals (Reaction 1.2). Formic acid is attacked by •OH and H• radicals to form •COOH, which reacts again with the water radicals to form $CO_2$ and $O_2$. $CO_2$ can react with H• to generate the •COOH radical.

Reactions 1.1 to 1.8 were transformed into a system of coupled ordinary differential equations (Eqs 1 and 2). Fig 2 displays the outcomes of computed solutions for the system of equations and depicts the degradation of formic acid under gamma radiation and the resultant production of $CO_2$ and $H_2$.

### 3.2 Formate at pH 9

The system involving formate at pH 9 comprises reactions that are different from and independent of the radiolysis of formic acid (Table 3). The primary attack is conducted by three water radicals (•OH, H•, and $e_{aq}^-$), forming the $•CO_2^-$ radical. This last radical serves as the basis for the secondary reactions that lead to the formation of oxalate ($^-OOC–COO^-$). In addition, oxalate reacts continuously with water radicals (Reactions 2.8 to 2.10).

The numerical model indicates the continuous decay of formate and the formation of the oxalate ion as the main radiolysis product (Fig 3). In this case, we also compared our numerical model with the experimental data of Horne et al. (2020) [42].

### 3.3 Formic acid and formate at pH 2 and high irradiation doses

Formic acid and formate maintain an acid-base equilibrium at pH 2 (Fig 1). In this scenario, the concentration ratio of formic acid to formate is 98:2, according to the equilibrium

**Table 2. Reaction mechanism for formic acid at pH 1.5 under a gamma radiation field.**

| Reaction | $k$ (s$^{-1}$) | Reference | React. No. |
|---|---|---|---|
| $H_2O \xrightarrow{\gamma-radiation} •OH, e_{aq}^-, H•, H_2O_2$ | - | [33] | {1.1} |
| $e_{aq}^- + H_{aq}^+ \xrightarrow{k0} H•$ | $k_0 = 2.6*10^{12}$ | [51] | {1.2} |
| $HCOOH + H• \xrightarrow{k1} •COOH + H_2$ | $k_1 = 4.4*10^5$ | [52] | {1.3} |
| $HCOOH + •OH \xrightarrow{k2} •COOH + H_2O$ | $k_2 = 1.4*10^8$ | [52] | {1.4} |
| $•COOH + H• \xrightarrow{k3} H_2 + CO_2$ | $k_3 = 2.1*10^8$ | [34–36] | {1.5} |
| $•COOH + •OH \xrightarrow{k4} CO_2 + H_2O$ | $k_4 = 2.6*10^9$ | [34–36] | {1.6} |
| $•COOH + H_2O_2 \xrightarrow{k5} CO_2 + •OH + H_2O$ | $k_5 = 5.0*10^7$ | [33, 42] | {1.7} |
| $H• + CO_2 \xrightarrow{k6} •COOH$ | $k_6 = 1.0*10^6$ | [53] | {1.8} |

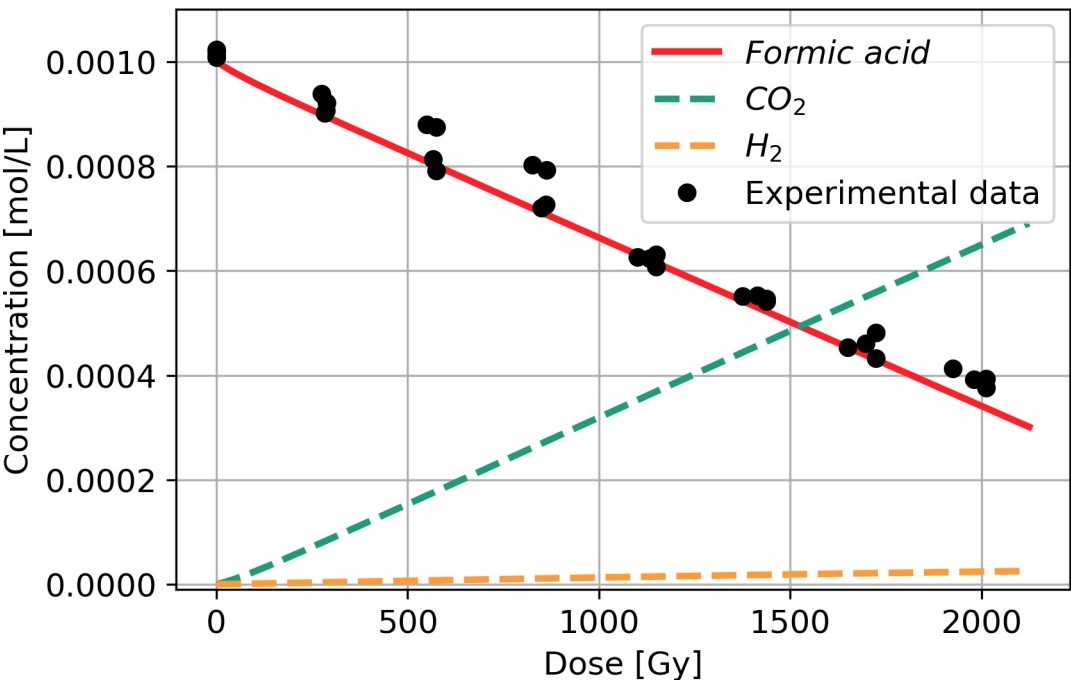

**Fig 2. Numerical modeling of the radiolysis of formic acid at pH 1.5 under oxygen-free conditions; the dashed lines represent the computed solutions for the resultant production of $CO_2$ and $H_2$.** The experimental data were extracted from the study by Horne et al. (2020), [42].

established in the studies by Joo et al. (2013, 2014) [43, 55]. Then, if the total molar concentration is 0.02 mol/L, the initial concentration of formic acid is 0.0196 mol/L, and that formate is 0.0004 mol/L.

In this case, the reaction system includes components from the preceding systems. It begins with Reactions 1.1 to 1.8 from the formic acid system and Reactions 2.1 and 2.2 from the formate system. High concentrations of formic acid under ionizing radiation result in the formation of larger molecules [41], (Table 4).

**Table 3. Reaction mechanism for the formate ion at pH 9 under a gamma radiation field.**

| Reaction | $k$ (s$^{-1}$) | Reference | React. No. |
|---|---|---|---|
| $H_2O \xrightarrow{\gamma-radiation} \bullet OH, e_{aq}^-, H\bullet, H_2O_2$ | | [33] | {1.1} |
| $HCOO^- + \bullet OH \xrightarrow{k7} \bullet CO_2^- + H_2O$ | $k_7 = 2.6*10^9$ | [52] | {2.1} |
| $HCOO^- + H\bullet \xrightarrow{k8} \bullet CO_2^- + H_2$ | $k_8 = 2.1*10^8$ | [52] | {2.2} |
| $HCOO^- + e_{aq}^- + H_2O \xrightarrow{k9} \bullet CO_2^- + H_2 + OH^-$ | $k_9 = 8.0*10^3$ | [52] | {2.3} |
| $2(\bullet CO_2^-) \xrightarrow{k10} OCOCOO^{2-}$ | $k_{10} = 7.6*10^8$ | [37, 42] | {2.4} |
| $OCOCOOO^{2-} \xrightarrow{k11} {}^-OOC-COO^-$ (oxalate) | $k_{11} = 1.0*10^9$ | [40, 42] | {2.5} |
| $\bullet CO_2^- + H\bullet \xrightarrow{k12} HCOO^-$ | $k_{12} = 9.0*10^9$ | [52] | {2.6} |
| $\bullet CO_2^- + e_{aq}^- + H_2O \xrightarrow{k13} HCOO^- + OH^-$ | $k_{13} = 9.0*10^9$ | [52] | {2.7} |
| $^-OOC-COO^- + \bullet OH \xrightarrow{k14} CO_2 + \bullet CO_2^- + OH^-$ | $k_{14} = 7.7*10^6$ | [54] | {2.8} |
| $^-OOC-COO^- + H\bullet \xrightarrow{k15} HOOC - C(OH)_2 + 2OH^-$ | $k_{15} = 1.0*10^4$ | [54] | {2.9} |
| $^-OOC-COO^- + e_{aq}^- \xrightarrow{k16} HOOC - C(OH)_2 + 3OH^-$ | $k_{16} = 4.8*10^7$ | [54] | {2.10} |

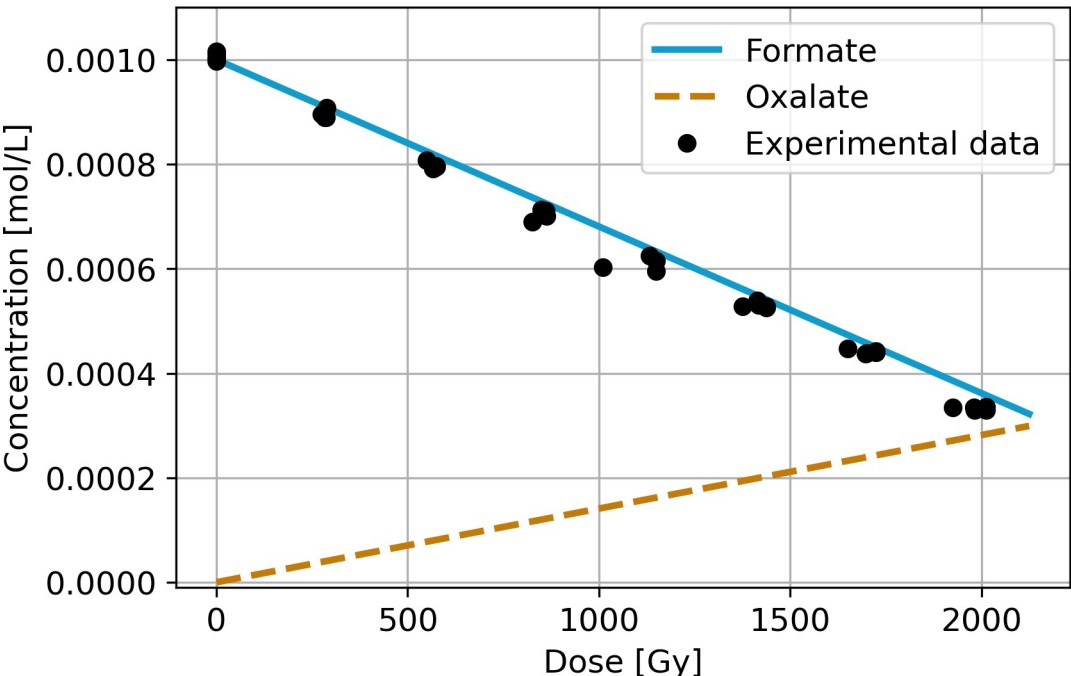

**Fig 3. Computed numerical model for formate under gamma radiation and the formation of oxalate.** The dots represent the experimental results of Horne et al. (2020), [42].

At pH 2, the available concentration of the solvated electron ($e_{aq}^-$) is low though not null [33]. This species reacts with formic acid to produce the OC•H radical, serving as an intermediate in the reaction that leads to the formation of acids and aldehydes. Formaldehyde (HCHO) and glyoxylic acid (CHOCOOH) have been reported as radiolysis products of concentrated formic acid solutions in previous studies [35].

At pH 2, •$CO_2^-$ radical reacts rapidly to produce $CO_2$, via either an electron transfer or disproportionation reaction [42, 56]. The rate constants of $k_{18}$, $k_{19}$, and $k_{20}$ are unknown; they correspond to secondary reactions between many unstable intermediates, making their determination difficult. The numerical stability window of the model caused by the variation of these unknown rate constants is between $1.0*10^2$ s$^{-1}$ and $1.0*10^8$ s$^{-1}$. The calculated solutions for formic acid and $CO_2$ show no significant changes in this interval. This stability window is wide because the reactions 3.2–3.4 are at the end part of the chain reaction, suggesting that the main reactions are controlled by the primary attack of the water radicals. The structure of the numerical model requires a number in each rate constant, so in order not to increase the

**Table 4. Complement of the reaction mechanism for formic acid and formate at pH 2 under a gamma radiation field.**

| Reaction | $k$ (s$^{-1}$) | Reference | React. No. |
|---|---|---|---|
| HCOOH + $e_{aq}^- \xrightarrow{k17}$ OC•H + OH$^-$ | $k_{17}$ = 1.9 *$10^5$ | [33] | {3.1} |
| HCOOH + OC•H $\xrightarrow{k18}$ HCHO + $H_2O$ (formaldehyde) | $k_{18}$ = unknown | [35] | {3.2} |
| 2(OC•H) $\xrightarrow{k19}$ HCHO + CO | $K_{19}$ = unknown | [41] | {3.3} |
| •COOH + OC•H $\xrightarrow{k20}$ CHOCOOH (glyoxylic acid) | $k_{20}$ = unknown | [36] | {3.4} |
| 2(• $CO_2^-$) + $H_2O \xrightarrow{k21}$ $CO_2$ + HCOO$^-$ | $k_{21}$ = 6.2*$10^8$ | [42] | {3.5} |

numerical stiffness of the system, we assumed a rate constant of $k = 1.0*10^4$ s$^{-1}$ for the reactions between the intermediates (Reactions 3.2 to 3.4). It is important to note that with this assumption we are not able to give information about the molecules that are produced in these reactions (formaldehyde or glyoxylic acid). If experimental information on these constants were obtained, and the constants were outside of the interval of stability, it would be necessary to resort to other numerical methods or to modify the one used to obtain a stable solution.

All reactions in the formic acid and formate system are expressed as a system of coupled ordinary differential equations (ODEs), according to Eqs 1 and 2:

$$\frac{dX_{H\bullet}(t)}{dt} = f_{H\bullet} + -k_1 X_{HCO_2H}(t)X_{H\bullet}(t) - k_3 X_{\bullet COOH}(t)X_{H\bullet}(t)$$
$$-k_6 X_{CO_2}(t)X_{H\bullet}(t) - k_8 X_{HCO_2^-}(t)X_{H\bullet}(t) \tag{3.1}$$

$$\frac{dX_{\bullet OH}(t)}{dt} = f_{OH\bullet} + k_5 X_{\bullet COOH}(t)X_{H_2O_2}(t)$$
$$-k_2 X_{HCO_2H}(t)X_{\bullet OH}(t) - k_4 X_{\bullet COOH}(t)X_{\bullet OH}(t)$$
$$-k_7 X_{HCO_2^-}(t)X_{\bullet OH}(t) \tag{3.2}$$

$$\frac{dX_{H_2O_2}(t)}{dt} = f_{H_2O_2} - k_5 X_{\bullet COOH}(t)X_{H_2O_2}(t) \tag{3.3}$$

$$\frac{dX_{H_{aq}^+}(t)}{dt} = f_{H_{aq}^-} - k_0 X_{e_{aq}^-}(t)X_{H_{aq}^-}(t) \tag{3.4}$$

$$\frac{dX_{e_{aq}^-}(t)}{dt} = f_{e_{aq}^+} - k_0 X_{e_{aq}^-}(t)X_{H_{aq}^-}(t) - k_{17}X_{HCO_2H}(t)X_{e_{aq}^-}(t) \tag{3.5}$$

$$\frac{dX_{HCO_2H}(t)}{dt} = -k_1 X_{HCO_2H}(t)X_{H\bullet}(t) - k_2 X_{HCO_2H}(t)X_{\bullet OH}(t)$$
$$-k_{17}X_{HCO_2H}(t)X_{e_{aq}^-}(t) - k_{18}X_{HCO_2H}(t)X_{OC\bullet H}(t) \tag{3.6}$$

$$\frac{dX_{\bullet COOH}(t)}{dt}$$
$$= +k_1 X_{HCO_2H}(t)X_{H\bullet}(t) + k_2 X_{HCO_2H}(t)X_{\bullet OH}(t) + k_6 X_{CO_2}(t)X_{H\bullet}(t)$$
$$-k_3 X_{\bullet COOH}(t)X_{H\bullet}(t) - k_4 X_{\bullet COOH}(t)X_{\bullet OH}(t) - k_5 X_{\bullet COOH}(t)X_{H_2O_2}(t)$$
$$-k_{20}X_{\bullet COOH}(t)X_{OC\bullet H}(t) \tag{3.7}$$

$$\frac{dX_{H_2}(t)}{dt} = +k_1 X_{HCO_2H}(t)X_{H\bullet}(t) + k_3 X_{\bullet COOH}(t)X_{H\bullet}(t) + k_8 X_{HCO_2^-}(t)X_{H\bullet}(t) \tag{3.8}$$

$$\frac{dX_{H_2O}(t)}{dt} = +k_2 X_{HCO_2H}(t)X_{\bullet OH}(t) + k_4 X_{\bullet COOH}(t)X_{\bullet OH}(t)$$
$$+k_5 X_{\bullet COOH}(t)X_{H_2O_2}(t) + k_7 X_{HCO_2^-}(t)X_{\bullet OH}(t)$$
$$+k_{18}X_{HCO_2H}(t)X_{OC\bullet H}(t) - k_{21}X_{\bullet CO_2^-}(t)X_{H_2O}(t) \tag{3.9}$$

$$\frac{dX_{CO_2}(t)}{dt} = +k_3 X_{\bullet COOH}(t)X_{H\bullet}(t) + k_4 X_{\bullet COOH}(t)X_{\bullet OH}(t) + k_5 X_{\bullet COOH}(t)X_{H_2O_2}(t)$$
$$+ k_{21}\left[X_{\bullet CO_2^-}(t)/2\right]X_{H_2O}(t) - k_6 X_{CO_2}(t)X_{H\bullet}(t) \tag{3.10}$$

$$\frac{dX_{HCOO^-}(t)}{dt} = -k_7 X_{HCO_2^-}(t)X_{\bullet OH}(t) - k_8 X_{HCO_2^-}(t)X_{H\bullet}(t)$$
$$+ k_{21}\left[X_{\bullet CO_2^-}(t)/2\right]X_{H_2O}(t) \tag{3.11}$$

$$\frac{dX_{\bullet CO_2^-}(t)}{dt} = +k_7 X_{HCO_2^-}(t)X_{\bullet OH}(t) + k_8 X_{HCO_2^-}(t)X_{H\bullet}(t) - k_{21}X_{\bullet CO_2^-}(t)X_{H_2O}(t) \tag{3.12}$$

$$\frac{dX_{OC\bullet H}(t)}{dt} = +k_{17}X_{HCO_2H}(t)X_{e_{aq}^-}(t) - k_{18}X_{HCO_2H}(t)X_{OC\bullet H}(t) - k_{19}X_{OC\bullet H}(t)$$
$$- k_{20}X_{\bullet COOH}(t)X_{OC\bullet H}(t) \tag{3.13}$$

$$\frac{dX_{OH^-}(t)}{dt} = +k_{17}X_{HCO_2H}(t)X_{e_{aq}^-}(t) \tag{3.14}$$

$$\frac{dX_{HCHO}(t)}{dt} = +k_{18}X_{HCO_2H}(t)X_{OC\bullet H}(t) + k_{19}X_{OC\bullet H}(t)/2 \tag{3.15}$$

$$\frac{dX_{CO}(t)}{dt} = +k_{19}X_{OC\bullet H}(t)/2 \tag{3.16}$$

$$\frac{dX_{CHOCOOH}(t)}{dt} = +k_{20}X_{\bullet COOH}(t)X_{OC\bullet H}(t) \tag{3.17}$$

The solutions of the coupled ODEs system of Eqs 3.1–3.17 are shown in Figs 4 and 5. Fig 4 shows the continuous decrease of the total molar concentration of formic acid and formate and the formation of $CO_2$ from 0 to 70 kGy. The total concentration of formic acid and formate are quantified together. $CO_2$ is the principal radiolysis product and is continuously formed.

The numerical model allows for the separation of the calculated solutions of the formic acid and formate equilibrium. In this case, it is displayed using both linear and $\log_{10}$ scales on the y-axis, with the latter enhancing the visualization of the molecules' behavior (Fig 5). Additionally, it is possible to follow the molar concentration of each molecule involved in the system.

## 3.4 Formic acid and formate at a pH of 3.75

Formic acid has an equivalence point ($pK_a$) at a pH of 3.75 [43]. Here, the concentration ratio of formic acid to the formate ion is 1:1. As a theoretical exercise, we developed a numerical model with a high concentration of formic acid at pH 3.75, to approximate the behavior of formic acid and the formate ion mixture under ionizing radiation. The aim of the exercise is to simulate another possible prebiotic environment. From a numerical point of view, we compute models at the two extremes of the pH gradient, when the systems have pure formic acid (section 3.1) and formate ion (section 3.2); and we have the elements to simulate the system when

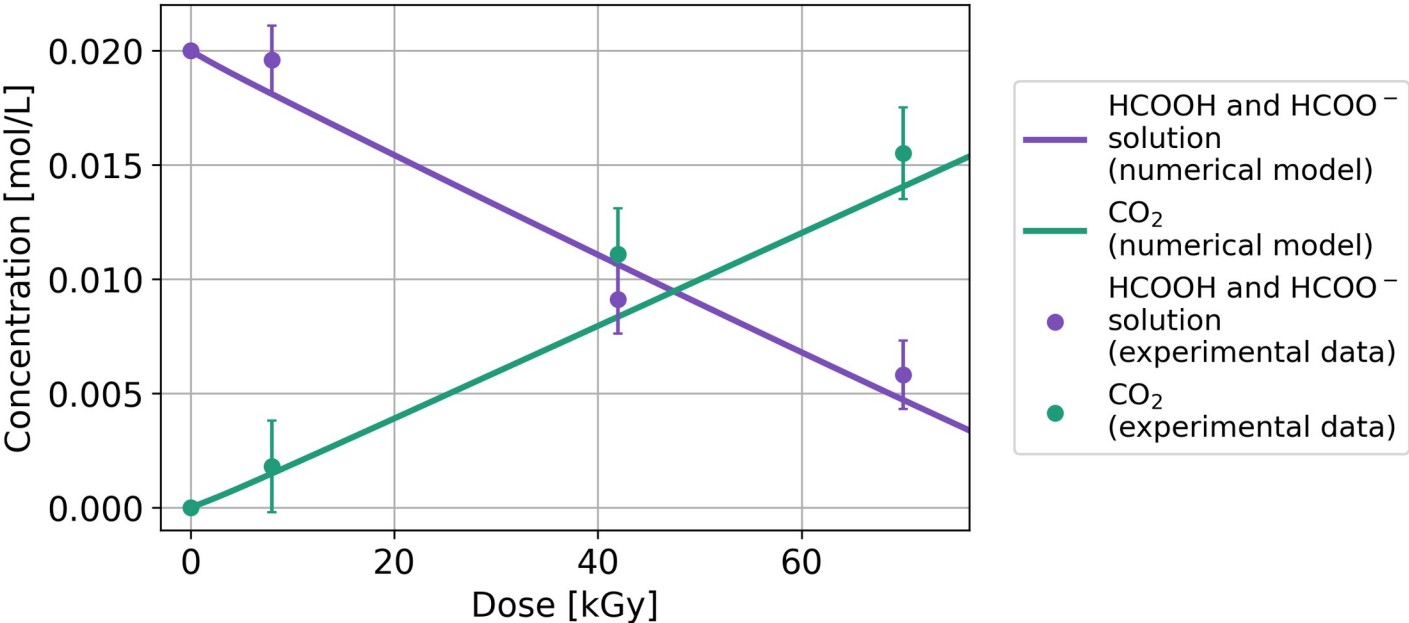

**Fig 4. The sum of formic acid and formate concentrations under gamma radiation from 0 to 70 kGy.** The lines represent the numerical model, and the dots correspond to the results obtained from the experimental setup used here.

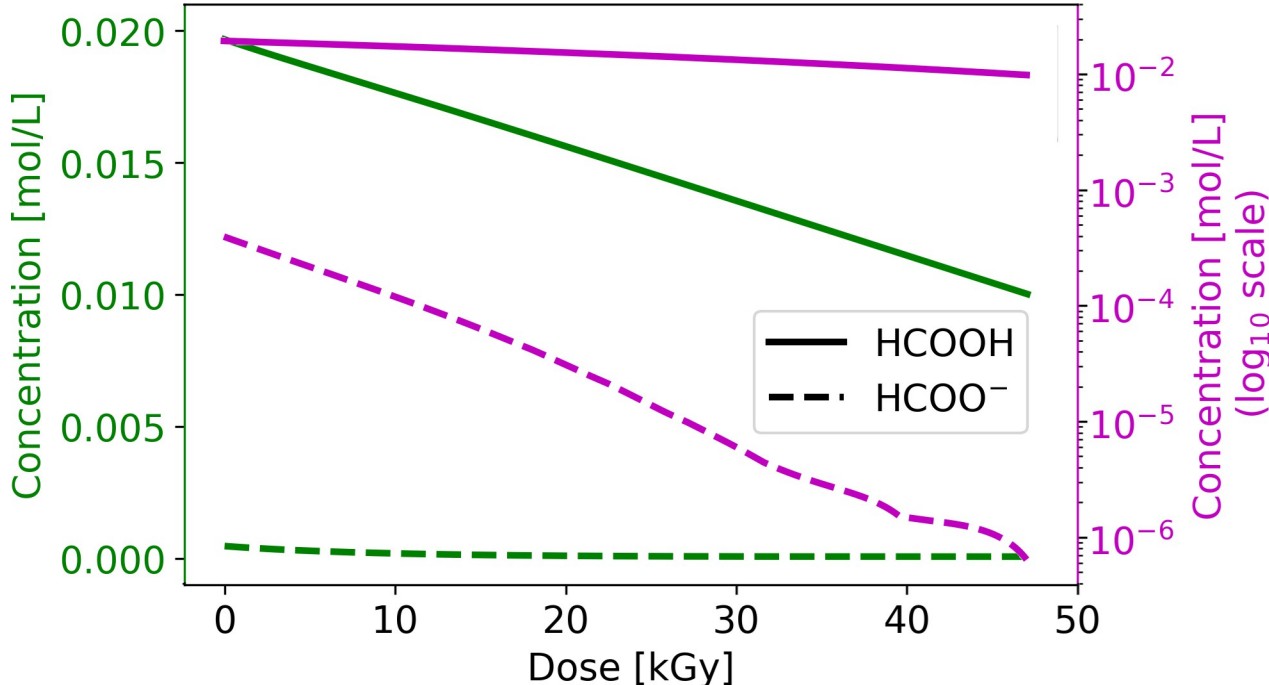

**Fig 5. Numerical model of the behavior of formic acid and formate under gamma radiation.** The left axis represents a linear scale, and the right axis involves the same computed solution on a $\log_{10}$ scale to enhance the visualization of formate decay.

**Table 5. Reaction mechanism for two formic acid radicals.**

| Reaction | $k$ (s$^{-1}$) | Reference | React. No. |
|---|---|---|---|
| $2(\bullet COOH) \xrightarrow{k23} (COOH)_2$ (oxalic acid) | $K_{23} = 4.0 \ast 10^5$ | Reaction [36] Constant [33] | {3.6} |

both molecules interact in the same environment at the same initial concentration. With this information we propose a system on the behavior of both molecules in conditions of an acid lake that their radiolysis generates a cluster of other molecules with biological importance.

In a 0.02 mol/L solution, the concentrations of both formic acid and formate are 0.01 mol/L. The reaction mechanism is the same as that of previously mentioned models, following Reactions 1.1 to 1.8, 2.1 and 2.2, and 3.1 to 3.4. Reaction 3.5 is substituted with Reaction 3.6 [33], as shown in Table 5, because above pH 3 the yield of $CO_2$ decreases, and it is replaced by oxalic acid [33, 57].

The model reveals the rapid degradation of formate followed by the degradation of formic acid. The available formate is depleted at 55 kGy, at which point the degradation rate of formic acid increases. The sum of formic acid and formate decay is a continuous function, and the major radiolysis product is oxalic acid. The model also shows a low formation of $CO_2$ (Fig 6A). For further theoretical insights, the initial conditions of this system at pH 3.75 were changed to an initial concentration of 0.001 mol/L for formic acid and the formate ion (1:1) and a total absorbed dose of 2kGy. This calculated solution was compared with the solutions of pure formic acid at pH 1.5 and pure formate at pH 9 (Fig 6B). The three computed solutions reveal the continuous decay of the molar concentrations under gamma irradiation with different degradation rates.

## 3.5 Statistical analysis

Finally, statistical analyses between the experimental data and the numerical calculations were performed (Table 6 and Fig 7). Each model yields an RMSE < 4%. The residual mean and residual standard deviation are < 3% for all the systems, except for the residual standard

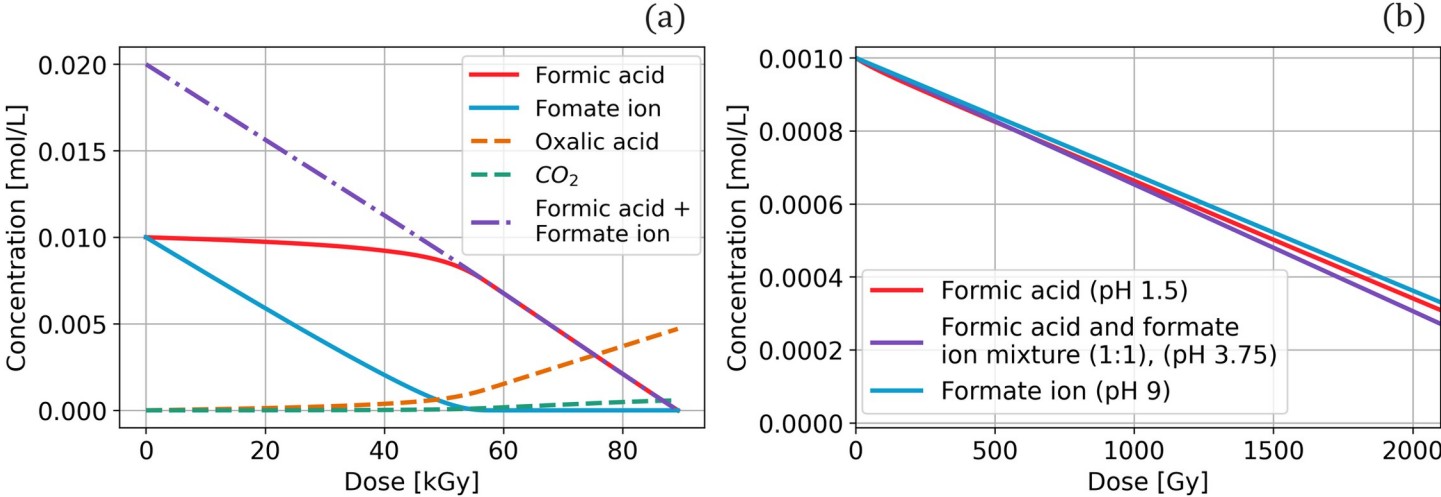

**Fig 6. Theoretical model of formic acid and formate under various pH conditions and total irradiation doses.** (a) Degradation of formic acid and the formate ion and the synthesis of oxalic acid and carbon dioxide at pH 3.75 under high irradiation doses from 0 to 80 kGy. (b) Formic acid degradation under various pH conditions and doses from 0 to 2 kGy.

**Table 6. Statistical analysis of the numerical solutions and the experimental results.**

| Molecule | RMSE | % | Residuals mean | % | Residuals SD | % |
|---|---|---|---|---|---|---|
| Formic acid | $3.71 \times 10^{-5}$ | 3.71 | $2.61 \times 10^{-5}$ | 2.61 | $2.68 \times 10^{-5}$ | 2.68 |
| Formate ion | $1.41 \times 10^{-5}$ | 1.41 | $1.96 \times 10^{-5}$ | 1.96 | $1.40 \times 10^{-5}$ | 1.40 |
| Formic & formate (pH 2) | $1.79 \times 10^{-3}$ | 8.96 | $3.61 \times 10^{-4}$ | 1.80 | $1.86 \times 10^{-3}$ | 9.31 |

deviation of the system at pH 2. The residual analysis reveals a close relationship between the calculated and experimental data from the three systems. The Q-Q plot suggests similar behavior for the laboratory experiments and the computational models (Fig 7).

## 4 Discussion

Each investigated system requires a different reaction mechanism because the equilibrium between formic acid and the formate ion depends on the pH [55]. Formic acid dominates at pH of 1.5. The exposure of pure formic acid to gamma radiation involves a relatively simple reaction mechanism (Reactions 1.1 to 1.8) and progressive decomposition. The numerical model based on this reaction system shows a continuous decrease in the molar concentration of formic acid from 0 to 2 kGy upon exposure to a $\gamma$-radiation field (Fig 1). $CO_2$ and $H_2$ are the main products of the radiolysis of formic acid [41]. The computed solution reveals the formation of these two molecules, with $CO_2$ constituting the major product, even with Reaction 1.8 regenerating the formic acid radical ($H \bullet + CO_2 \xrightarrow{k6} \bullet COOH$). The statistical calculations support the correspondence between the numerical model and the experimental data of Horne et al. (2020), with $R^2 > 0.99$.

The aqueous formate solution involves a more complex reaction mechanism than formic acid. At pH 9 and a concentration of 0.001 mol/L, oxalate ($^-OOC–COO^-$) is the major radiolysis product [57]. Oxalate is formed by the interaction of two formate radicals ($\bullet CO_2^-$); it also interacts with water radicals (Reactions 2.8 to 2.10), leading to the formation of other larger molecules. This is the final stage of the reaction mechanism in our numerical model since this

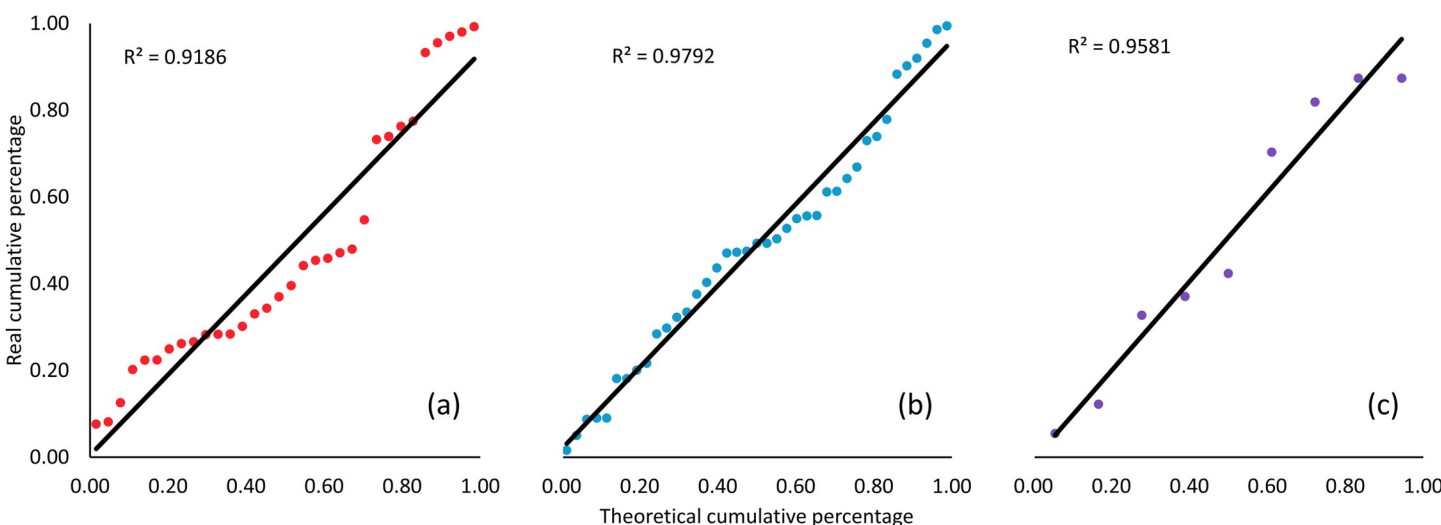

**Fig 7.** Quantile-quantile plot of residual data from the three models and experimental results: (a) formic acid at pH 1.5, (b) formate ion at pH 9, and (c) formic acid and formate at pH 2.

study aimed to model the first steps of radiolysis and the main products. The calculated results closely correspond to the experimental data, with $R^2 > 0.97$.

The system at pH 2, comprising the formic acid and formate equilibrium, a high-concentration solution, and at high radiation dose, behaves differently. These variables generate more reactions in the system (Reactions 3.1 to 3.5) and yield more products, such as formaldehyde, glyoxylic acid, and acetaldehyde. The experimental results indicates that the decomposition of formic acid is constant between 0 and 70 kGy and that $CO_2$ is the major radiolysis product. Computational solutions for the coupled system of Eqs 3.1–3.17 can reproduce the general behavior of the experimental results, and monitoring all the molecules of this system, including the formation of each product, is possible. The solution can be observed separately for formic acid and formate, allowing the detailed tracking of each molecule (Fig 6). The calculated solution and experimental data yield an $R^2$ value$> 0.96$. Notably, all statistical analyses yield a high correlation between the experimental data and the calculated solutions. $R^2 > 0.96$ in each model, and the RMSE is less than 4% for pure formic acid and pure formate and less than 9% for the system at pH 2. The residual mean is less than 3% in each system, validating the model.

We developed the model of formic acid at pH 3.75 to obtain a theoretical approximation of the behavior of formic acid and the formate ion under ionizing radiation. The numerical model (Fig 6A) agrees with the observed decrease in $CO_2$ yield and increase in oxalic acid concentration, indicating that the main radiolysis product is oxalic acid, and that $CO_2$ is formed at a low concentration. Furthermore, it shows that the concentration of formate decreases faster than that of formic acid; this is attributed to the substitution of Reaction 3.5,

$$2\left(\bullet CO_2^-\right) + H_2O \xrightarrow{k21} CO_2 + HCOO^-,$$ with Reaction 3.6, $2(\bullet COOH) \xrightarrow{k23} (COOH)_2$, restricting the regeneration of formic acid [33]. This model shows faster decay under ionizing radiation than the model at pH 2 (Figs 4 and 6A). It is possible to separate the numerical solutions of formic acid and the formate ion, and Fig 6A indicates that the formate ion decays faster than formic acid. This can be explained by the attack of H• radical on both molecules because the mixture displays competition reactions. The rate constant for the reaction of formic acid with H• is $k_1 = 4.4*10^5$ s$^{-1}$ (Reaction 1.3, $HCOOH + H \bullet \xrightarrow{k1} \bullet COOH + H_2$), and that for the reaction between the formate ion and H• is $k_8 = 2.1*10^8$ s$^{-1}$ (Reaction 2.2,

$$HCOO^- + H \bullet \xrightarrow{k8} \bullet CO_2^- + H_2).$$ The reactions display a difference in the order of magnitude of the rate constant, with the formate ion exhibiting a faster reaction rate than formic acid. Similarly, the reaction of the •OH radical with the formate ion ($k_7 = 2.6*10^9$ s$^{-1}$) is one order of magnitude faster than that with formic acid ($k_2 = 1.4*10^8$ s$^{-1}$), as shown in Reactions 2.1 and 1.4, respectively. Formic acid and formate at pH 3.75 decay linearly and completely at 89 kGy, whereas at pH 2, 75% of the total molar concentration decays at 70 kGy.

Experimental work for this model, with the radiolysis and identification of all products, can be conducted in future studies. A variation of this model, changing the initial concentration of the total formic acid and formate to 0.001 mol/L, was compared with the calculated solutions of the models involving pure formic acid and pure formate (Fig 6B). The solution of the pure formate ion is the most stable of these systems, probably because the attack of $e_{aq}^-$ on the formate ion (Reaction 2.3, $k_9 = 8.0*10^3$ s$^{-1}$) is slower than the attack of the H• radical on formic acid (Reaction 1.3, $k_1 = 4.4*10^5$ s$^{-1}$). The H• radical and the $e_{aq}^-$ are in an acid-base equilibrium, where H• dominates in an acidic medium and $e_{aq}^-$ dominates in a basic medium. The mixture of formic acid and formate at pH 3.75 decays faster than pure formic acid and pure formate. This can be attributed to the competition kinetics of the reactions because this system involves more molecules reacting to gamma radiation, and there are fewer reactions allowing the regeneration of initial molecules or reaction intermediates.

The dose constant ($\hat{k}$) for each system was also calculated, as show in Fig 6B. The dose constant is a function of the dose and concentration, expresses the reaction kinetics as a pseudo-first-order equation as a first approximation, and is descriptive only. It is an empirical indicator of the resistance of molecules to radiation [58, 59], as given by Eq 4:

$$ln\left(\frac{C}{C_0}\right) = -\hat{k}D \Rightarrow \hat{k} = -ln\left(\frac{C}{C_0}\right)/D \tag{4}$$

where $C$ is the concentration, $C_0$ is the initial concentration, $D$ is the applied dose (Gy), and $\hat{k}$ is the dose constant ($Gy^{-1}$). The obtained constants with Eq 4 have the same order of magnitude and reinforce the abovementioned point, that the 1:1 mixture of formic acid and formate degrades faster than the pure formic acid and pure formate ion systems (Fig 8).

Monitoring every molecule and free radical in a chemical system is essential to understanding chemical evolution processes. One of the study goals was to elucidate the development and implications of each molecule with a possible role in prebiotic chemistry processes. This was based on the importance of formic acid and formate and their radiolysis products, such as other carboxylic acids and amino acids, in biological systems. The agreement between the numerical models and the experimental results supports the proposed reaction mechanisms for the behavior of each system. The numerical models allow one to generate hypotheses

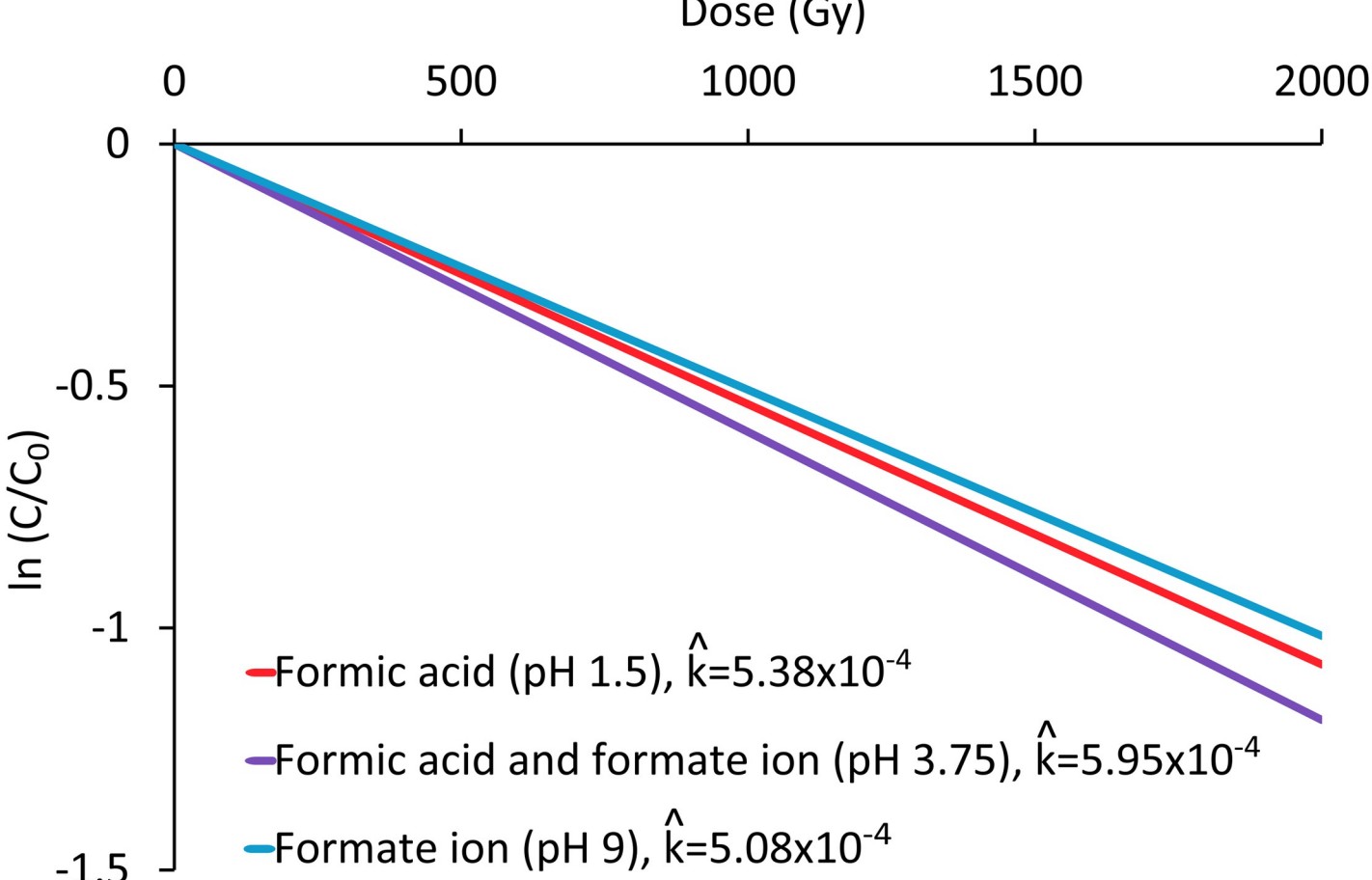

**Fig 8. Dose constant ($\hat{k}$) for 0.001 mol/L solutions of formic acid at pH 1.5, the 1:1 mixture of formic acid and formate at pH 3.75, and the formate ion at pH 9.**

regarding the behavior of formic acid and its radiolysis products in various systems with high ionizing radiation fields and pH variations (from acidic to basic media), such as shallow lakes, certain zones of hydrothermal springs, comets, exoplanets, etc.

One should note that the method given herein to model the behavior of organic molecules in aqueous solution under ionizing radiation fields is a simplified approximation of the phenomenon, based only on chemical kinetics and not considering the diffusion coefficients of the molecules in aqueous media.

## 5 Conclusions

The behavior of formic acid and formate under gamma radiation changes with the pH conditions and molar concentrations used. Our numerical model can reproduce this behavior under such different conditions, where the molecules in each system can undergo numerous reactions. Since these models are based on reaction kinetics, identifying the dominant reaction mechanism is necessary to provide robust solutions. The equilibrium model between formic acid and formate involves more variables than the models for pure formic acid and pure formate. The computed solutions show a strong relationship with the experimental data, and the high statistical significance aids in validating each system and reaction mechanism. The models presented in this work can help elucidate the role played by formic acid and its products in a variety of possible prebiotic environments. Our proposed model thus extends the potential molecules and environments that can be simulated relevant to prebiotic chemistry.

## Acknowledgments

A.P.A. acknowledge to CONAHCyT (CVU 929149) for the PhD fellowship and to Posgrado en Ciencias de la Tierra—UNAM for the support of the PhD studies. The authors acknowledge the PAPIIT IN114122 and CONAHCyT 319118 projects. The authors express their gratitude to C. Camargo-Raya for her technical help in the Laboratorio de Evolución Química, ICN-UNAM; and to Dr. B. Leal-Acevedo, Phys. J. Gutiérrez-Romero and M. Sc. M. J. Rodriguez Albarrán for the irradiation of the samples, and E. Palacios-Boneta, M. Cruz-Villafañe and J. Rangel-Gutiérrez for technical support. Finally, the authors would like to thank the anonymous reviewers for their comments and suggestions, which helped to improve the paper. This work was conducted at and supported by the Instituto de Ciencias Nucleares, UNAM.

## Author Contributions

**Conceptualization:** Alejandro Paredes-Arriaga, Diego Frias, Ana Leonor Rivera, Sergio Ramos-Bernal, Alicia Negrón-Mendoza.

**Formal analysis:** Alejandro Paredes-Arriaga, Diego Frias, Ana Leonor Rivera, Alicia Negrón-Mendoza.

**Funding acquisition:** Sergio Ramos-Bernal, Alicia Negrón-Mendoza.

**Investigation:** Alejandro Paredes-Arriaga, Anayelly López-Islas, Guadalupe Cordero-Tercero.

**Methodology:** Alejandro Paredes-Arriaga, Anayelly López-Islas.

**Resources:** Sergio Ramos-Bernal, Alicia Negrón-Mendoza.

**Supervision:** Diego Frias, Guadalupe Cordero-Tercero, Sergio Ramos-Bernal, Alicia Negrón-Mendoza.

**Writing – original draft:** Alejandro Paredes-Arriaga, Alicia Negrón-Mendoza.

**Writing – review & editing:** Diego Frias, Ana Leonor Rivera, Guadalupe Cordero-Tercero.

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
