## [Decision Letter · Decision Letter 0]

8 Oct 2024

PONE-D-24-33362Numerical modeling of formic acid and formate ion behavior under gamma radiation: Implications on prebiotic chemistryPLOS ONE

Dear Dr. Paredes-Arriaga,

Thank you for submitting your manuscript to PLOS ONE. After careful consideration, we feel that it has merit but does not fully meet PLOS ONE’s publication criteria as it currently stands. Therefore, we invite you to submit a revised version of the manuscript that addresses the points raised during the review process. Please submit your revised manuscript by Nov 11 2024 11:59PM. If you will need more time than this to complete your revisions, please reply to this message or contact the journal office at plosone@plos.org. Please include the following items when submitting your revised manuscript:A rebuttal letter that responds to each point raised by the academic editor and reviewer(s). You should upload this letter as a separate file labeled 'Response to Reviewers'.A marked-up copy of your manuscript that highlights changes made to the original version. You should upload this as a separate file labeled 'Revised Manuscript with Track Changes'.An unmarked version of your revised paper without tracked changes. You should upload this as a separate file labeled 'Manuscript'.

We look forward to receiving your revised manuscript.

Kind regards,

Parag A. Deshpande

Academic Editor

PLOS ONE

Journal Requirements:

Reviewers' comments:

Reviewer's Responses to Questions

**Comments to the Author**

1. Is the manuscript technically sound, and do the data support the conclusions?

Reviewer #1: Yes

Reviewer #2: Yes

2. Has the statistical analysis been performed appropriately and rigorously? 

Reviewer #1: Yes

Reviewer #2: Yes

3. Have the authors made all data underlying the findings in their manuscript fully available?

Reviewer #1: No

Reviewer #2: Yes

4. Is the manuscript presented in an intelligible fashion and written in standard English?

Reviewer #1: Yes

Reviewer #2: Yes

5. Review Comments to the Author

Reviewer #1: Authors report a combined experimental/computational work on the formic acid and formate ion behaviour under gamma radiation. The topic looks properly inserted into the existing literature. The English form is fluent, the computational setup and theoretical background are properly presented, the results are discussed in a detailed and clear manner. The work is promising to elucidate important reaction mechanisms involving the formic acid and the formate ion, which are at the basis of prebiotic environment. As the experimental setup falls outside my expertise, I cannot express an opinion about its reliability.

Therefore, I recommend the publication of the manuscript after the positive opinion of an expert in the presented experimental part presented and after the following minor revision:

1) lines 140, 141, "Each coupled nonlinear differential equation system was solved using our Python (3.7.9) code...": Authors should report the full code as supporting information together with description on how to run the code, in order to allow the reproducibility of the results. Indeed, the latter is a mandatory requirement in any scientific work.

Reviewer #2: The authors present numerical kinetics models for several pH and irradiation dose profiles for formic acid/formate, using, mostly, literature rate constants, but also providing new data for one case. I think the work meets PLOS-ONE criteria of being technically rigorous and meeting scientific standards for contributions to the literature. Agreement between the various models and experiment is very good. I think it could be published once the following points have been taken care of.

In the order of the ms.:

1) The title indicates only numerical modelling but not the experimental work. I'd suggest modifying it, perhaps something along the lines of "Numerical modeling vs experiment for formic acid..."

2) on p 5, line 97, I'd change kinetics chemistry to chemical kinetics

3) On p. 6, some rationale is needed for the choices of these variables. They may make sense for a radiation-chemistry audience but the general PLOS ONE reader would need some indication of why those particular values are appropriate.

4) On p. 7 just under Eq (1) something is wrong with " k_i,j^(i) is the rate constant for species j and k produced by the species i". There is no sub- or super-script k in Eq (10).

5) Also, I think some general explanation of Eq (1) is needed.

6) On p. 7, I think a reference is needed for Eq 2, perhaps to a radiochemistry textbook?

7) On p. 15, the rationale for choosing k=10^4 seems weak to me. If the real values are greater than 10^4 then, yes, abrupt changes could occur, but they could be real. What would happen if k is less than 10^4?

8)on p. 18, what is the point of this "theoretical exercise". The rationale for substituting oxalic acid for CO2 is only given much later in the ms. Please bring this explanation up to the beginning of the Section so that the reader knows why you are doing this.

9) "at a pKa" isn't correct. pKa is a property of the acid, pH is the controlled quantity so pH should appear in the title and text, with an explanation early on that the chosen pH is equivalent to the pKa of formic acid.

6. PLOS authors have the option to publish the peer review history of their article (what does this mean?). If published, this will include your full peer review and any attached files.

Reviewer #1: No

Reviewer #2: No

---

## [Author Response · Author response to Decision Letter 0]

8 Nov 2024

Reviewer #1

1) lines 140, 141, "Each coupled nonlinear differential equation system was solved using our Python (3.7.9) code...": Authors should report the full code as supporting information together with description on how to run the code, in order to allow the reproducibility of the results. Indeed, the latter is a mandatory requirement in any scientific work.

We have attached the link and cited the corresponding GitHub repository. Here, you can find the general algorithm to solve the system of differential equations based on a reaction mechanism and the full code of the formic acid model with the description and the instructions to run it.

Reviewer #2: 

1) The title indicates only numerical modelling but not the experimental work. I'd suggest modifying it, perhaps something along the lines of "Numerical modeling vs experiment for formic acid..."

The text was changed as suggested.

2) on p 5, line 97, I'd change kinetics chemistry to chemical kinetics

The text was changed as suggested

3) On p. 6, some rationale is needed for the choices of these variables. They may make sense for a radiation-chemistry audience but the general PLOS ONE reader would need some indication of why those particular values are appropriate.

We add the corresponding justification in the beginning of the Methodology section. 

4) On p. 7 just under Eq (1) something is wrong with " k_i,j^(i) is the rate constant for species j and k produced by the species i". There is no sub- or super-script k in Eq (10).

That is true, it was a mistake. We made corrections in the text.

5) Also, I think some general explanation of Eq (1) is needed.

We expand the explanation of Equation 1 below it.

6) On p. 7, I think a reference is needed for Eq 2, perhaps to a radiochemistry textbook?

We add the corresponding references.

7) On p. 15, the rationale for choosing k=10^4 seems weak to me. If the real values are greater than 10^4 then, yes, abrupt changes could occur, but they could be real. What would happen if k is less than 10^4?

We extend the rationale for choosing the rate constant k=104, talking about numerical window stability in this part of the reaction chain, and deepen about the scenario if the k would be less or more than the stability interval.

8)on p. 18, what is the point of this "theoretical exercise". The rationale for substituting oxalic acid for CO2 is only given much later in the ms. Please bring this explanation up to the beginning of the Section so that the reader knows why you are doing this.

The text has been changed as suggested, and we extend the explanation about the objective of this theoretical model.

9) "at a pKa" isn't correct. pKa is a property of the acid, pH is the controlled quantity so pH should appear in the title and text, with an explanation early on that the chosen pH is equivalent to the pKa of formic acid.

Thank you so much for that observation. Title and text have been changed as suggested.

---

## [Decision Letter · Decision Letter 1]

26 Nov 2024

Numerical modeling vs experiment of formic acid and formate ion behavior under gamma radiation at several pH values: Implications on prebiotic chemistry

PONE-D-24-33362R1

Dear Dr. Paredes-Arriaga,

We’re pleased to inform you that your manuscript has been judged scientifically suitable for publication and will be formally accepted for publication once it meets all outstanding technical requirements.

Kind regards,

Parag A. Deshpande

Academic Editor

PLOS ONE

Additional Editor Comments (optional):

Reviewers' comments:

Reviewer's Responses to Questions

**Comments to the Author**

1. If the authors have adequately addressed your comments raised in a previous round of review and you feel that this manuscript is now acceptable for publication, you may indicate that here to bypass the “Comments to the Author” section, enter your conflict of interest statement in the “Confidential to Editor” section, and submit your "Accept" recommendation.

Reviewer #2: All comments have been addressed

2. Is the manuscript technically sound, and do the data support the conclusions?

Reviewer #2: Yes

3. Has the statistical analysis been performed appropriately and rigorously? 

Reviewer #2: N/A

4. Have the authors made all data underlying the findings in their manuscript fully available?

Reviewer #2: Yes

5. Is the manuscript presented in an intelligible fashion and written in standard English?

Reviewer #2: Yes

6. Review Comments to the Author

Reviewer #2: (No Response)

7. PLOS authors have the option to publish the peer review history of their article (what does this mean?). If published, this will include your full peer review and any attached files.

Reviewer #2: No

---

## [Editor Report · Acceptance letter]

2 Dec 2024

PONE-D-24-33362R1 

PLOS ONE

Dear Dr. Paredes-Arriaga, 

I'm pleased to inform you that your manuscript has been deemed suitable for publication in PLOS ONE. Congratulations! Your manuscript is now being handed over to our production team.

Kind regards, 

on behalf of

Dr. Parag A. Deshpande 

Academic Editor

PLOS ONE